# Transgenerational deep sequencing revealed hypermethylation of hippocampal mGluR1 gene with altered mRNA expression of mGluR5 and mGluR3 associated with behavioral changes in Sprague Dawley rats with history of prolonged febrile seizure

**Oluwole Ojo Alese** [ID]*, **Musa V. Mabandla**

Department of Human Physiology, College of Health Sciences, University of Kwazulu-Natal, Durban, South Africa

* alese44@yahoo.com

## Abstract

The impact of febrile seizure has been shown to transcend immediate generation with the alteration of glutamatergic pathway being implicated. However, transgenerational effects of this neurological disorder particularly prolonged febrile seizure (PFS) on neurobehavioral study and methylation profile is unknown. We therefore hypothesized that transgenerational impact of prolonged febrile seizure is dependent on methylation of hippocampal mGluR1 gene. Prolonged febrile seizure was induced on post-natal day (PND) 14, by injecting lipo-polysaccharide (LPS; 217μg/kg *ip*) and kainic acid (KA; 1.83 mg/kg *ip*). Sucrose preference test (SPT) and Forced swim test (FST) were carried out in the first generation ($F_0$) of animals at PND37 and PND60. The $F_0$ rats were decapitated at PND 14, 37 and 60 which corresponded to childhood, adolescent and adulthood respectively and their hippocampal tissue collected. The second generation ($F_1$) rats were obtained by mating $F_0$ generation at PND 60 across different groups, $F_1$ rats were subjected to SPT and FST test on PND 37 only. Decapitation of $F_1$ rats and collection of hippocampal tissues were done on PND 14 and 37. Assessment of mGluR5 and mGluR3 mRNA was done with PCR while mGluR1 methylation profile was assessed with the Quantitative MassARRAY analysis. Results showed that PFS significantly leads to decreased sucrose consumption in the SPT and increased immobility time in the FST in both generations of rats. It also leads to significant decrease in mGluR5 mRNA expression with a resultant increased expression of mGluR3 mRNA expression and hypermethylation of mGluR1 gene across both generations of rats. This study suggested that PFS led to behavioral changes which could be transmitted on to the next generation in rats.

**Data Availability Statement:** All relevant data are within the paper and its Supporting Information files.

**Funding:** This research was funded by the College of Health Sciences grant University of Kwazulu-natal Durban (85017060-425085). The funder had no role in study design, data collection and analysis, decision to publish, or preparation of the manuscript.

**Competing interests:** The authors have declared that no competing interests exist.

# Introduction

Febrile seizure is the most prevalent seizure type, affecting 3% to 5% of infants between the ages of 3months and 5years [1,2]. It is caused by infections (including respiratory tract infection, gastroenteritis and otitis media) that trigger the immune system leading to inflammatory responses with a subsequent release of macrophages, neutrophils and pro-inflammatory cytokines [3,4]. Sequel to these, the infections cause systemic fever that subsequently leads to aberrant neuronal excitability in febrile seizure [5,6,7]. However, one important factor that lead to the characterized aberrant neuronal excitability in febrile seizure, is neurotransmitter imbalance between glutamate and γ aminobutyric acid (GABA) [8].

Glutamate is the predominant excitatory neurotransmitter in the brain that has been associated with aberrant signaling and pathology of febrile seizure [7]. Furthermore, glutamate mediates its excitatory effects through two receptors that are ionotropic and metabotropic. Ionotropic glutamate receptors are dependent on ionic gradients while metabotropic glutamate receptors facilitate their action by coupling to secondary G-protein messengers [9]. Among the metabotropic glutamate receptors, the activation of metabotropic glutamate receptor 1 and 5 (mGluR1, mGluR5) enhances presynaptic glutamate release, thus modulating glutamatergic signaling in seizure [7]. Also, the metabotropic glutamate receptor 3 (mGluR3) regulates glutamate release on the presynaptic membrane and glia cell [10]. More importantly, previous studies have revealed up-regulation of the expression of mGluR1 on astrocytes in epilepsy and seizure [7,11,12]. Likewise hypermethylation of mGluR1 gene has been reported in malignant melanoma in mouse [13] and in the hippocampus of prenatal stressed rats offspring with depression [14,15].

Previous studies showed that different genetic alterations may be linked to abnormal behavior and functioning of the developing brain [16,17]. Behavioral and emotional changes have been linked to an early life exposure to unfavorable environmental factors which can be transferred to the offspring [18,19,20]. It has been demonstrated that stress in childhood may result in emotional disturbance in adulthood which could also be transferred across generations [21]. Early childhood PFS exposure in rats has been shown to cause memory deficit in adulthood that can be epigenetically transferred to their offspring [22]. Epigenetic alteration has been attributed to these environmental variations and as such, transferrable to their offspring [23]. Besides, epigenetics is the chromatin network modifications which lead to alteration of gene expression leaving the DNA sequence intact [24]. Epigenetic modifications have been reported to affect regulatory processes involved in learning and memory, synaptic plasticity and central nervous system development [25]. However, such epigenetic modification of mGluR1 in prolonged febrile seizure even across generation of rats is yet to be determined. Hence, we designed the present study to investigate the transgenerational effects of PFS through hippocampal mGluR5and mGluR3 mRNA expression. Also, we examined the transgenerational effects of PFS on neurobehavioral assessments and the methylation profile of mGluR1 gene in Sprague Dawley rats.

# Materials and methods

## Animals

All experimental procedures and animals used were treated in accordance with the approved guideline by the Animal Ethics Research Committee of the University of KwaZulu-Natal (AREC/042/016D).The adult rats were obtained from the Biomedical Resource Center of the University of KwaZulu-Natal, where they were housed under standard laboratory conditions with a 12 h light/dark cycle (lights on at 06h: 00). The rats had access to food and water *ad*

*libitum*. A total of 64 pups (56 males and 8 females) obtained from mating female and male adult Sprague- Dawley rats were used in the experiments. The sample size was set according to previous studies where the statistical power was shown [26].

## Experimental procedure

On post-natal day (PND) 14, prolonged febrile seizures (PFS) was induced in 32 pups (28 males and 4 females) by injecting lipopolysaccharide (LPS, 217 µg/kg) intraperitoneally (i.p) followed by an injection of kainic acid (KA, 1.83 mg/kg *i.p*) at 2h 30min later [27]. LPS had been demonstrated to reliably increase body temperature [28]. Temperature was monitored with the use of a digital thermometer before LPS injection and thereafter every 30 min until rats were given KA [29]. The remaining animals were grouped as control and injected with a corresponding volume of normal saline (SAL). Animals were monitored for one hour following the injection of KA and seizure severity was observed as follows: stage 0: no change in behavior; stage 1: chewing; stage 2: gazing and head nodding; stage 3: unilateral forelimb clonus, twitching and scratching; stage 4: rearing with bilateral forelimb clonus; stage 5: widespread muscle spasms, rearing with bilateral forelimb clonus and falling back [30]. All rats induced had at least stage 4 convulsions for a minimum of 30min duration, indicating that the convulsion were visibly obvious to the observer as rearing with bilateral forelimb clonus which is consistent with earlier observation [30]. In the first generation ($F_0$), 8 males (4 PFS and 4 SAL) rats were decapitated at PND 14 1h after the induction of PFS. The remaining pups were then returned to their mothers and weaned on PND 21.Two sets of 20 male rats from the $F_0$ generation (10 PFS and 10 SAL in each set) were decapitated after the neurobehavioral test at PND 37 and PND 60 respectively. This timelines of PND 14, 37 and 60 corresponds to the life span of the Sprague-Dawley rats from childhood, through adolescent to adulthood respectively. On PND 60, the second generation ($F_1$) rats were obtained by mating the remaining 16 rats (8males and 8 females) according to Table 1 below. A total of 16 rats comprising of 4offsprings from each mating group were sacrificed on PND 14 across the various groups while 28 rats were subjected to neurobehavioral test and decapitated on PND 37 across the various groups in $F_1$.

## Sucrose preference test (SPT)

The SPT was conducted over a 24 h period to check for anhedonia in the animals [31]. A day prior to the test, (PND 34 and 57)in $F_0$ rats and PND 34 in $F_1$ generation, the rats were weighed and placed in separate cages for a 24 h training period. The training phase consists of placing two bottles of known volume of tap water on opposite sides of the cage [32]. After 24 h (PND 35 and 58), one bottle of water was replaced with another containing 5% sucrose solution. For the test phase, the two bottles (one containing tap water and the other 5% sucrose solution) were placed on the left and right sides respectively. After 24 h, the bottles were weighed to determine consumption in grams (converted to ml). The sucrose preference was calculated as the volume of sucrose drank over the total volume of fluid consumed (sucrose

**Table 1. Table showing the first generation ($F_0$) animal grouping for mating.**

| $F_0$ 1 | $F_0$ 2 | $F_0$ 3 | $F_0$ 4 |
|---|---|---|---|
| SALm + SALf | PFSm + SALf | PFS f + SALm | PFSf + PFSm |

Key: SALm; Saline adult male, SALf; Saline adult female, PFSm; Prolonged febrile seizure adult male, PFSf; Prolonged febrile seizure adult female.

+ water) x 100. A decreased amount of sucrose solution consumed is suggestive of anhedonia hence depressive-like behavior [31].

## Forced swim test (FST)

The forced swim test assesses the despair behaviour in animals [33]. On PND 36 and 59 in $F_0$ rats and PND 36 in $F_1$ rats, each rat was placed in a plexiglass cylinder (50 H X30 D in cm) filled with water (24 ± 0.5 $^{o}$C) to a level of 30 cm for 15 min as a pre-test [33]. This pre-test is required to enable quick adoption of an immobile posture by rats on the test day hence an obvious despair behaviour which is expected in depressed rats. The test session (5 min) was conducted on PND 37 and 60in $F_0$ rats and PND 37 in $F_1$ rats. The immobile time that the rats spent by floating on water with the head slightly immersed and/or minor movements to maintain the head above water after the 1st min was video-recorded and documented for each animal [34].

## Decapitation

All rats were decapitated using a sharp guillotine. Hippocampal tissues were immediately dissected, weighed and placed in Eppendorf tubes before snap freezing in liquid nitrogen. The tissues were stored in a -80˚C bio-freezer until the day of analysis.

## Real-time PCR

Twenty (20) micrograms of hippocampal tissue were weighed, homogenized and suspended in 300 μl of RNA lysis buffer (Zymo Research, USA). Thereafter, RNA isolation was carried out as instructed in manufacturer's protocol (ZR RNA MiniPrep$^{TM}$, USA). Purification of RNA isolates was assessed using a NanoDrop (Thermo Scientific USA). Purity of 1.8–2.01 was recommended for use in the construction of cDNA. The cDNA synthesis was carried out using the iScript$^{TM}$cDNA Synthesis Kit (BioRad, South Africa) and run through the Thermocycler according to instructions stipulated in the protocol. The Fast start SYBR green kit (Roche Diagnostics, USA) was used in accordance with the manufacturer's guidelines. The mGluR5 (Grm5) and mGluR3 (Grm3) primers used to assay genes were designed by Inqaba Biotech (Pretoria, South Africa) as shown in Table 2. Primer sequences were reconstituted in RNA nuclease free water according to the manufacturer's protocol and were added to a master mix comprising of SYBR green dye, nuclease free $H_2O$ and $MgCl_2$. Thereafter, cDNA was added to the strips and run in the Light cycler 480 (Roche USA) at optimized conditions as instructed in the manufacturer's protocol. Experiments were performed in triplicate, GAPDH gene was used as the house keeping control and Data were analyzed by comparing C(q) values of the PFS samples to SAL and both normalized to GAPDH using the 2-ΔΔCq comparative method [35]

## Quantitative MassARRAY analysis of gene methylation status

**DNA extraction.** Genomic DNA was extracted from the hippocampal tissue using a DNeasy® Blood and Tissue Kit according to the manufacturer's instructions (Qiagen, Hilden,

**Table 2. Nucleotide sequence of forward and reverse primers for real-time PCR.**

| Target mRNA bases | Forward primer | Reverse primer |
| --- | --- | --- |
| GAPDH | GCCAAAAGGGTCATCATCTCCGC | GGATGACCTGCCCACAGCCTTG |
| Grm5 | TCCAGCAGCCTAGTCAACCT | CAGATTTTCCGTTGGAGCTT |
| Grm3 | CGCTCTCCTAATCTCCCTCTGG | CTCCTCTTCTCTTATCAGG |

Germany). Twenty-five mg of hippocampal tissue of each animal were weighed and placed in a microcentrifuge tube. Then, 180µl of buffer ATL and 20µl of proteinase K was added into the tubes, incubated in a water bath at 56 °C and vortexed intermittently until the tissue were lysed completely. Subsequently, 200 µl of AL buffer was added and incubated at 56 °C for 10min. Thereafter, 200 µl of 100% ethanol was added and mixed thoroughly. The mixture was then pipetted into the DNeasy Mini spin column placed in 2ml collection tube and centrifuged at 8000 rpm for 1 min (HERMLE Labortechnik GmbH, Germany), the flow-through and the collection tubes were discarded. The Mini spin column was again placed into a new 2 ml collection tube with 500 µl of buffer AW1 added via the column and centrifuge at 8000 rpm for 1min, the flow-through and the collection tubes were discarded. Subsequently, spin column was placed in a new collection tube,500 µl of buffer AW2was added and the mixture was spun for 3min at 14000 rpm with the flow-through and the collection tubes discarded. To elute the DNA, the spin column was placed in 2 ml collection tube and 200 µl of buffer AE was added, incubated at room temperature for 1min and centrifuged for 1min at 8000rpm.This elution step was repeated to enhance the yield. The concentration and purity of the DNA were determined by absorbances at 260–280 nm with a ratio 1.7–2.0 by NanoDropTM 1000 spectrophotometer (Thermo Scientific, Wilmington, USA).

**DNA sodium bisulfite conversion.** Sodium bisulfate modification of the extracted DNA was performed using an EZ DNA Methylation Kit[TM] according to the manufacturer's instructions (Zymo Research, Orange, CA, USA). Five microliter of M-Dilution buffer was added to each DNA sample of 500ng, adjusted to a total volume to 50µl with distilled water and incubated at 37˚C for 15min. Then, 100µl of CT conversion reagent was added to each sample and mixed properly. The samples were incubated in the thermocycler for 20 cycles at 95˚C for 30s and 50˚C for 15 min, followed by a final holding step at 4˚C.A volume of 400 µl of M-Binding buffer was added to a Zymo-Spin™ IC Column placed in a collection tube, this was then followed by loading the sample from the previous preparation above and mixing thoroughly by inverting the columns severally. The preparation was centrifuged at 8000rpm for 30s and the flow-through discarded, this was followed by adding 200µl of M-Wash buffer to the column and then centrifuged at full speed for 30s. Subsequently, 200µl of M-Desulphonation buffer was added to the column and incubated at room temperature (20–30˚C) for 20 min. It was there after centrifuged at full speed for 30s;200µl of M-Wash buffer was added to the column and centrifuged at 14000rpm for 30s. The above step was repeated and the columns were placed into a 1.5ml microcentrifuge tube. Furthermore, 20 µl of pre-warm M-Elution buffer was added directly to the column matrix, incubated at room temperature for 1 min and then centrifuged for 30s at full speed to elute the DNA.

**Quantitative mass ARRAY analysis of gene methylation status.** The Sequenom MassARRAY platform was used for the quantitative methylation analysis of the upstream sequence of mGluR1 gene promoter. The region analysed and the CpG sites of mGluR1 promoter are shown in Fig 1. The primers used in this study were designed by Inqaba Biotech (Pretoria, South Africa) (forward primer AGGAAGAGAGTTGTTAGGTATTTTGGGTAAAATGG and reverse primer CAGTAATACGACTCACTATAGGGAGAAGGCTAAACCCAAAAATTTAAATA CAATTCC). The PCR mixture (10 ng bisulfite-treated DNA, 25 mMdNTP, 5U/µL of PCR enzyme and a 1µM mixture of forward and reverse primers) was pre-heated for 4 min at 94˚C and then incubated for 45 cycles of 94˚C for 20s, 56˚C for 30 s, and 72˚C for 60s, followed by 72˚C for 3 min. Two microliters of SAP mix containing 1.7µl$H_2O$ and 0.3µl (1.7 U) of shrimp alkaline phosphatase (Sequenom) was added to digest redundant dNTPs with the following program: 37˚C for 20 min, 85˚C for 5 min, then maintained at 4˚C. Five microliters of T Cleavage Transcription/RNase Cocktail, including 0.89 µl 5x T7 polymerase buffer, 0.22 µl T cleavage mix, 3.14 m Mdithiothreitol (DDT) 0.22µl, 22 U of T7 RNA and DNA Polymerase

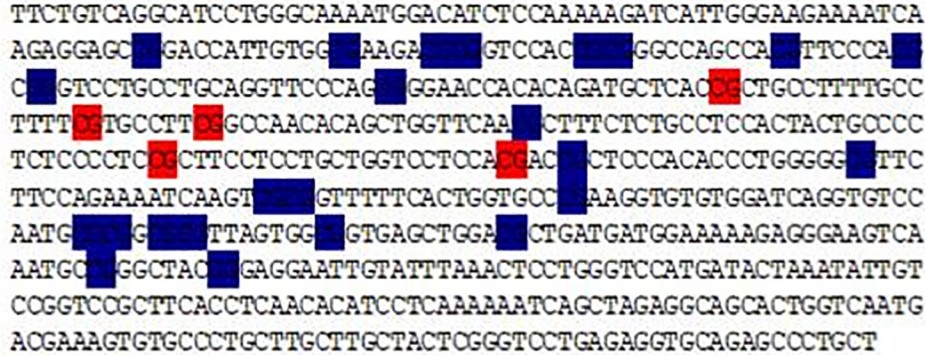

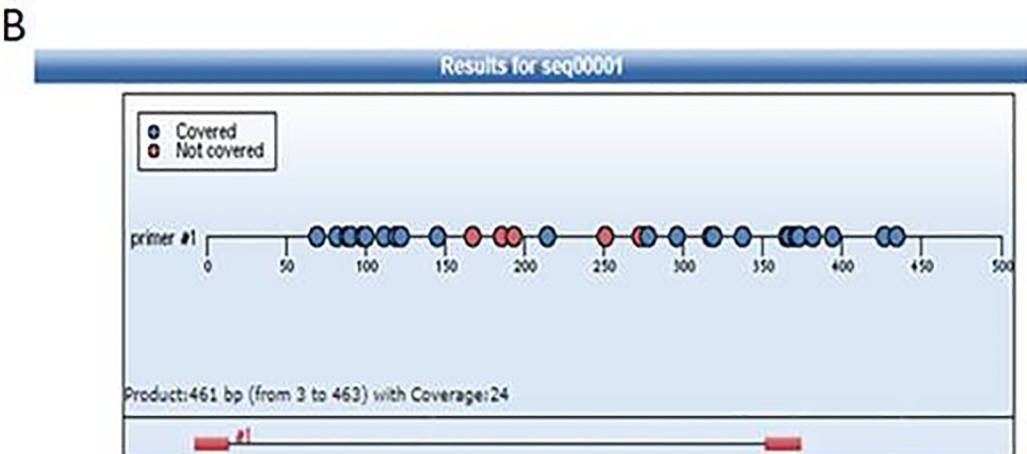

**Fig 1. Upstream sequence of mGluR1 core promoter region.** (**Fig 1A**) The chart is the analysis result of Methyl Primer Express; red pillars stand for CpG sites (**Fig 1B**) Another analysis in the same region; blue dots represent valid CpG sites while red dots which are invalid cannot be detected in MassARRAY, valid dots were numbered.

0.40μ, 0.09 mg/ml RNase A, and 2 μl of product of the PCR/SAP reactions were mixed and incubated under the following conditions: 37˚C for 3 h in vitro transcription and RNase A digestion. Then the mixture was further diluted with $H_2O$ to 27 μl, purified with CLEAN resin (Sequenom) and robotically dispensed onto silicon chips preloaded with matrix (Spectro-CHIP; Sequenom). The spectra and the methylation values of matrix-associated laser desorption/ ionization time-of-flight mass spectrometries (Sequenom) were collected and analyzed using Epityper software (version 1.0; Sequenom). Inapplicable readings and their corresponding sites were eliminated from analysis. The methylation level was expressed as the percentage of methylated cytosines over the total number of methylated and unmethylated cytosines.

## Statistical analysis

The data were analyzed using the Graph Pad Prism (version 5). Results were presented as mean ± SEM. The Shapiro-Wilk test was used to test the distribution of the data and determine

whether the data were parametric and non-parametric. Student t-tests were used to compare the genes and neurobehavioral study at different timelines among the first generation ($F_0$). One-way analysis of variance (ANOVA) with Newman-Keuls post hoc test was used to compare the genes and neurobehavioral characteristics across different groups in the second generation of animals ($F_1$). A $p < 0.05$ was considered statistically significant in all analysis.

# Results

## Sucrose preference test

Sucrose consumption was measured among groups of rats in both $F_0$ and $F_1$ generation. In the $F_0$ generation, rats with PFS showed significantly low sucrose preference when compared to SAL rats on PND 37 (p = 0.0004, t = 4.371) and PND 60 (p = 0.0006, t = 4.371) respectively (Fig 2A and 2B). In Fig 2C, there was a significant difference in PFS effect on the $F_1$ generation as offspring from both male and female PFS parents (PFSf + PFSm) on PND 37 showed significant low sucrose preference [F (3, 24) = 8.926, p = 0.0004] when compared to offspring from other groups of parents.

## Immobility during the forced swim test

The immobile time spent during the forced swim test was assessed in SAL and PFS rats. Fig 3A showed that a prolonged febrile seizure effect was evident at PND 37 in the $F_0$ generation during the forced swim test as PFS animals spent significant higher immobile time (p<0.0001, t = 5.230) when compared to SAL animals. In Fig 3B, on PND 60, PFS rats in the $F_0$ generation spent significantly more immobile time when compared to SAL rats(p<0.0001, t = 5.886). There was no significant PFS effect across groups in the $F_1$ generation of rats [F (3, 24) = 0.2206, p = 0.8811] (Fig 3C).

## Metabotropic glutamate receptor 5 (mGluR5) expression in hippocampal tissue

There was a PFS effect in the $F_0$ and $F_1$ generations of rats as shown in Fig 4. On PND 14, PFS animals showed a significantly decreased hippocampal mGluR5 mRNA expression when compared to the saline groups (p = 0.0272, t = 2.905, Fig 4A). Also, on PND 37, rats with PFS showed significantly decreased expression of mGluR5 mRNA when compared to SAL (p = 0.0126, t = 3.515,Fig 4B). Similarly, this was noticed on PND 60 in the $F_0$ animals, PFS rats showed decreased expression of mGluR5 mRNA in the hippocampus when compared to SAL animals (p = 0.0186, t = 3.199,Fig 4C). As shown in Fig 4D, there was a significant PFS effect on hippocampal mGluR5 mRNA transferred from the parents to their offspring [F (3, 12) = 5.904, p = 0.0103]. Offspring from adult males with PFS and saline females (PFSm + SALf) showed significantly decreased mGluR5 mRNA expression when compared to offspring from adult male and female saline rats (SALm+SALf) (p<0.05) (Fig 4D). Offspring from adult male with PFS and saline female (PFSm+SALf) showed significantly low mGluR5 expression when compared to offspring from adult male saline and female PFS animals (PFSf + SALm) (p<0.05) (Fig 4D). Offspring from adult male and female rats with PFS (PFSm + PFSf) showed significantly low mGluR5 expression when compared to offspring from adult male and female saline rats (SALm+SALf) (p<0.05) (Fig 4D). Similar observations were made among offspring from adult male and female with PFS as they showed significantly low mGluR5 expression when compared to offspring from adult male saline and female PFS rats (PFSf+SALm) (p<0.05) (Fig 4D).

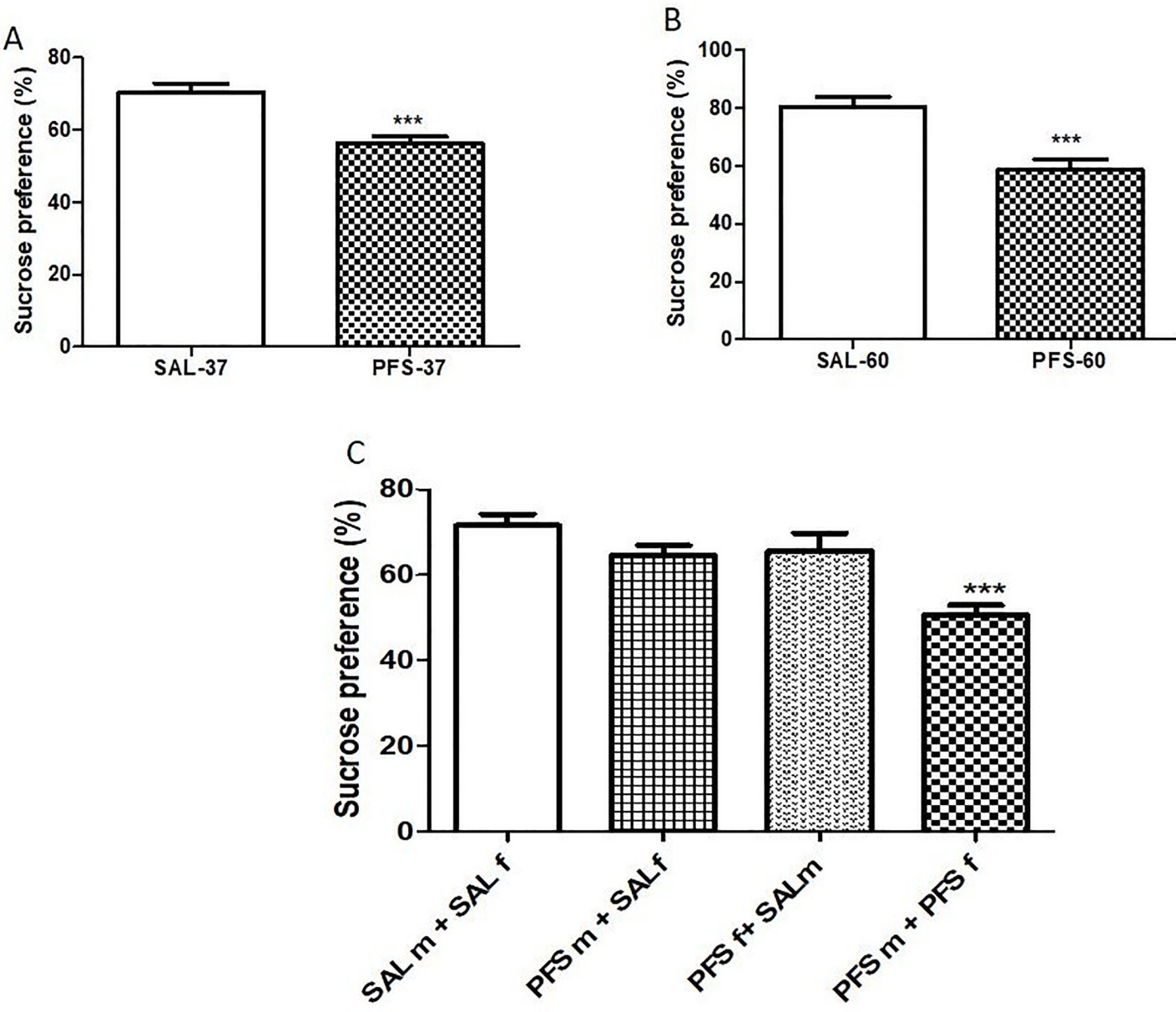

**Fig 2. Effects of history of PFS on sucrose preference test. Fig** 2A represents $F_0$ generation on PND 37***(SAL vs. PFS, p = 0.0004). Results expressed as mean ± SEM (n = 10/group). **Fig 2B**: $F_0$ rats at PND 60***(SAL vs. PFS, p = 0.0006). Results expressed as mean ± SEM (n = 10/group). **Fig 2C** showed the effect of PFS on $F_1$ generation at PND 37 ***(PFSm+PFSf vs. SALm+SALf, PFSm+SALf, PFSf+SALm, p = 0.0004). Results expressed as mean ± SEM (n = 7/group).

### Metabotropic glutamate receptor 3 (mGluR3) expression in hippocampal tissue

There was a PFS effect in the $F_0$ generation of rats as shown in Fig 5. In Fig 5A among the $F_0$ generation on PND 14, PFS animals had significantly high hippocampal mGluR3 mRNA expression when compared to the SAL groups (p = 0.0343, t = 2.728, Fig 5A). Also, on PND 37 rats with PFS showed significant high expression of mGluR3 mRNA when compared to SAL (p = 0.0008, t = 6.203, Fig 5B). Similar effects were seen on PND 60 in the $F_0$ generation PFS rats as there was increased expression of mGluR3 mRNA in the hippocampus when compared

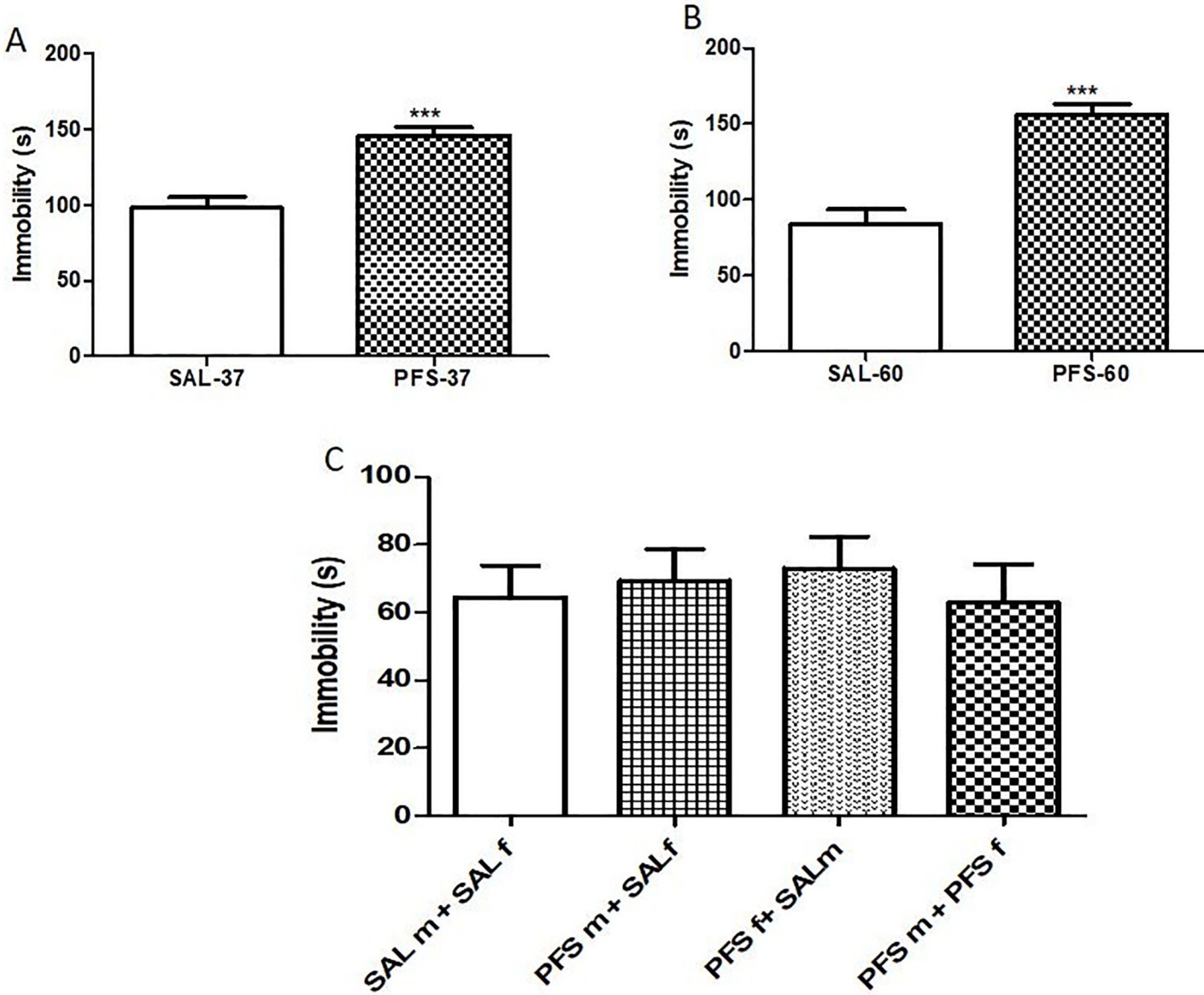

**Fig 3. Effects of PFS on the amount of time rats exhibited immobility during the forced swim test. Fig 3A** represents $F_0$ generation on PND 37***(SAL vs. PFS, p<0.0001). Results expressed as mean ± SEM (n = 10/group). **Fig 3B**: $F_0$ rats on PND 60 ***(SAL vs. PFS, p<0.0001). Results expressed as mean ± SEM (n = 10/group). **Fig 3C** showed the effect of PFS on $F_1$ generation. On PND 37 no significant PFS effect across groups in the $F_1$ generation of rats [F (3, 24) = 0.2206, p = 0.8811]. Results expressed as mean ± SEM (n = 7/group).

to SAL animals (p = 0.0207, t = 3.114,Fig 5C). As shown in Fig 5D, there was a PFS effect on hippocampal mGluR3 gene transferred from the parents to their offspring [F (3, 12) = 11, 24, p = 0.0008]. Offspring from adult male with PFS and saline female rats (PFSm+SALf) showed significantly higher mGluR3 mRNA expression when compared to offspring from adult male and female saline rats (SALm+SALf) (p<0.05) (Fig 5D). Offspring from adult male and female saline (SALm+SALf) showed significantly low mGluR3 mRNA expression when compared to offspring form adult male saline and female PFS (PFSf+SALm) (p<0.05) (Fig 5D). Offspring from adult male and female with PFS (PFSm+PFSf) showed significantly high mGluR3 mRNA

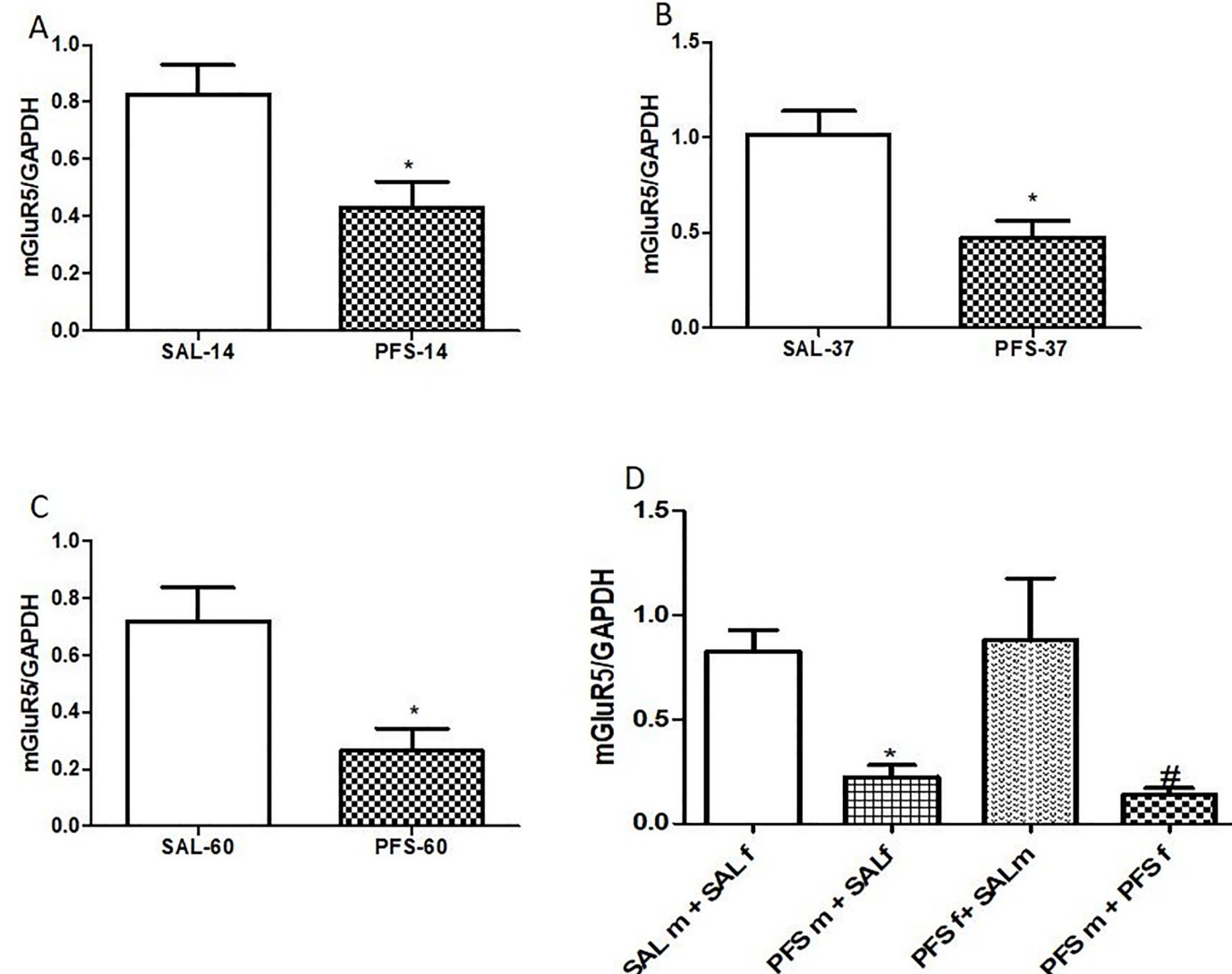

**Fig 4. Effects of PFS on the expression of hippocampal mGluR5 mRNA. Fig 4A** represent $F_0$ generation on PND 14*(SAL vs. PFS, p = 0.0272), **Fig 4B**: $F_0$ rats at PND 37 *(SAL vs. PFS, p = 0.0126), **Fig 4C**: $F_0$ at PND 60 *(SAL vs. PFS, p = 0.0188). **Fig 4D** showed the effect of PFS on mGluR5 mRNA in the $F_1$ generation on PND 37* (PFSm+SALfvs. SALm+SALf, PFSf+SALm p = 0.0103), #(PFSm+PFSf vs. SALm+SALf, PFSf+SALm, p = 0.0103). Results expressed as mean ± SEM (n = 4/group).

expression when compared to offspring form adult male and female saline rats (SALm+SALf) (p<0.05) (Fig 5D).

## Metabotropic glutamate receptor 1 (mGluR1) methylation level in hippocampal tissue

On PND 14, as shown in Fig 6A, PFS led to a significant decreased level of methylation among the $F_0$ generation when compared to the saline groups at CpG_4 and CpG_5 (p<0.05).It was also observed that among them, there were significant higher levels of methylation at CpG-11 and CpG_15 when compared to the saline group (p< 0.05).No significant difference was found at other CpG units. The methylation level of the CpG regions was also evaluated on PND 37(Fig 6B). Fig 6B shows that in the $F_0$ generation there were significant increase in the

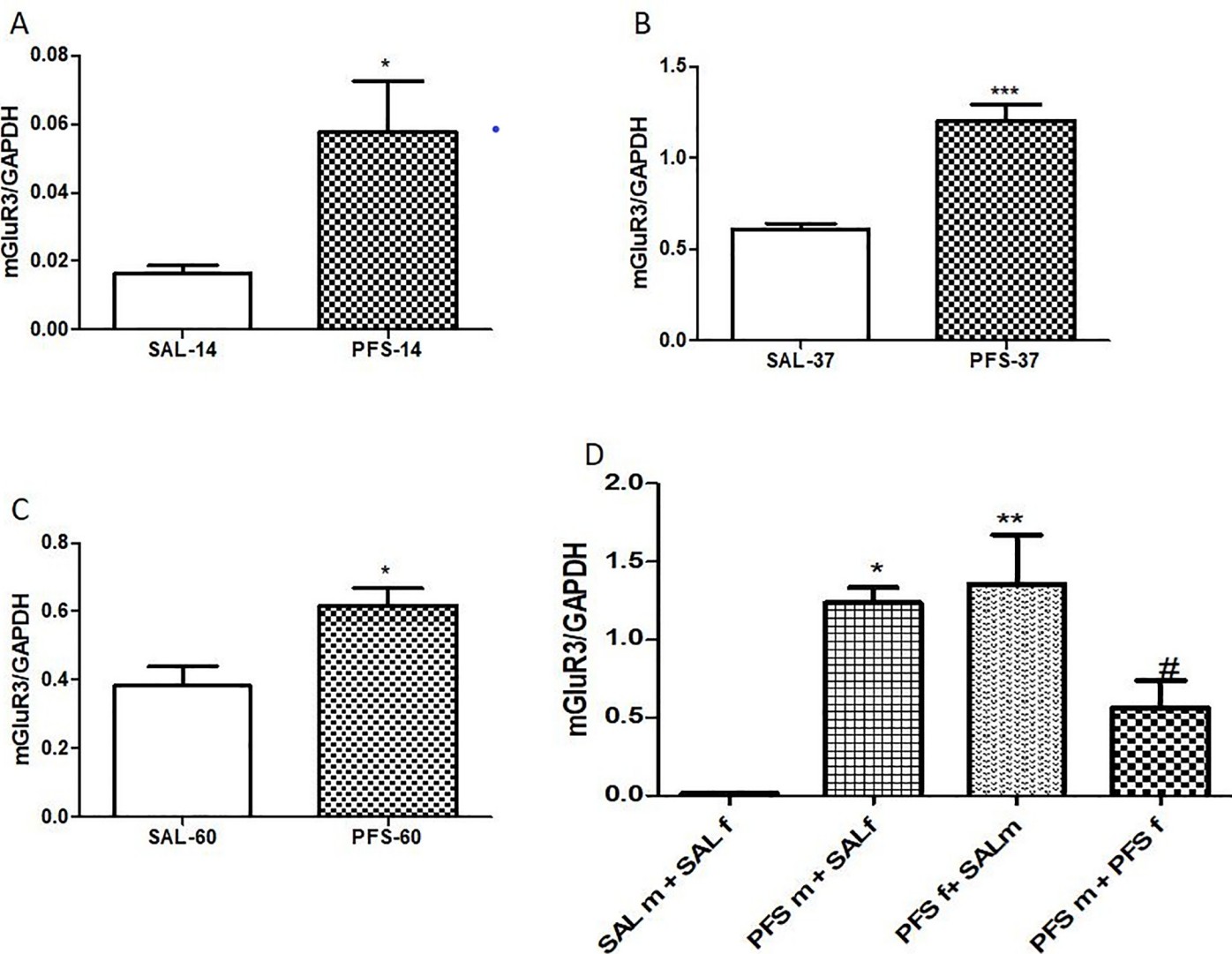

**Fig 5. Effects of PFS on the expression of hippocampal mGluR3mRNA. Fig 5A** represents $F_0$ generation at PND 14*(SAL vs. PFS, p = 0.0343), **Fig 5B**: $F_0$ rats at PND 37 ***(SAL vs. PFS, p = 0.0008), **Fig 5C** $F_0$ at PND 60 *(SAL vs. PFS, p = 0.0207). **Fig 5D** showed the effect of PFS on mGluR3 gene in the $F_1$ generation at PND 37 * (PFSm+SALf vs. SALm+SALf), ** (PFSf+SALmvs. SALm+SALf), #(PFSm+PFSf vs. SALm+SALf). Results expressed as mean ± SEM (n = 4/group).

methylation level of animals with prolonged febrile seizure when compared to saline at CpG_2, CpG_9, CpG_12, CpG_13 and CpG_14 (p< 0.05). There are no significant differences seen at other CpG units. Fig 6C shows that there were significant increases in the methylation level of animals in the $F_0$ generation on PND 60 with prolonged febrile seizure when compared to saline at CpG_9, with p< 0.05 on PND 60. There was no significant difference at other CpG units. Fig 6D shows the comparison between the different groups of offspring in the $F_1$ generation at the various CpG levels on PND 14. The results however showed that the methylation levels were significantly higher [F (3, 12) = 3,284, p = 0.0584]in offspring of PFSm +PFSf at CpG_8 when compared to offspring from both saline parents(SALm+SALf), but no significant difference in methylation level between the other CpG units in the $F_1$ generation of rats.

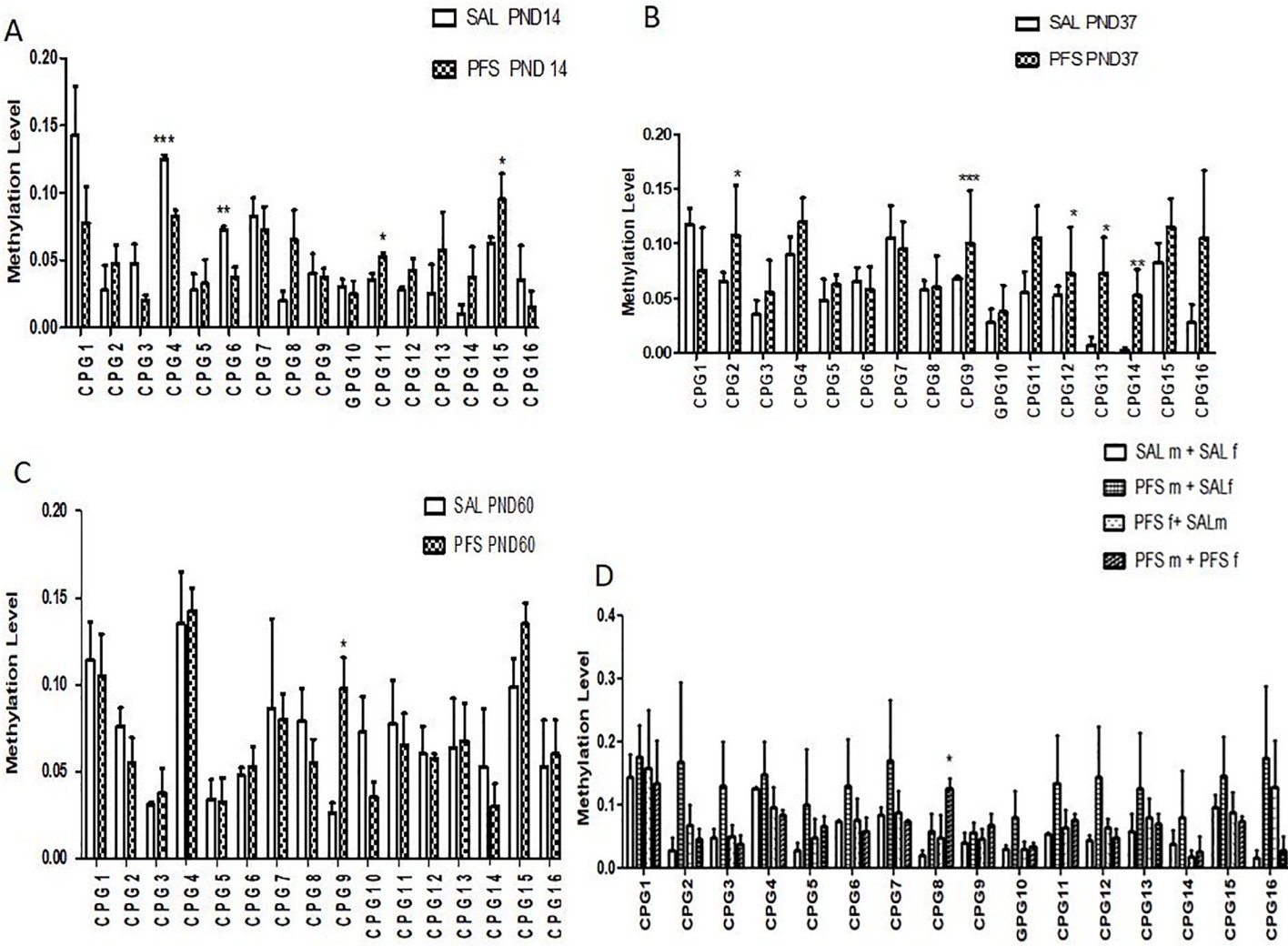

**Fig 6.** **Fig 6A:** Effects of PFS on the $F_0$ generation hippocampal methylation level of mGlR1 gene on PND 14 rats. Data are expressed as mean ±SEM (n = 4/group).*** (CpG _ 4 p = 0.0003), ** (CpG_6 p = 0.004), *(GpG_11 p = 0.02) and *(CpG_15 p = 0.04). **Fig 6B:** Effects of PFS on the hippocampal methylation level of mGlR1 gene in rats on PND 37 in the $F_0$ generation. Data are expressed as mean±SEM (n = 4/group).*** (CpG _ 9 p = 0.0005), ** (CpG_14 p = 0.004), *(GpG_2 p = 0.04) and *(CpG_12; 13 p = 0.03). **Fig 6C:** Effects of PFS on the $F_0$ generation hippocampal methylation level of mGlR1 gene in rats on PND 60. Data are expressed as mean ±SEM (n = 4/group).* (CpG _ 9 p<0.0005). **Fig 6D:** Effects of PFS on the hippocampal methylation level of mGluR1 gene on $F_1$ generation rats. *(PFSm+PFSf vs. SALm+SALf, CpG _8 p = 0.0584). Data are expressed as mean ±SEM (n = 4/group).

## Discussion

In this study, we examined the effect of neonatal PFS on SPT and FST in rats from adolescence to adulthood in the $F_0$ generation and at the adolescent age in the $F_1$ generation. We also looked at the effects of PFS on hippocampal mGluR1gene, mGlur5 mRNA and mGluR3mRNA at different timelines across both $F_0$ and $F_1$ generations of rats. We observed that animals with PFS developed anhendonia by consuming less sucrose solution in the SPT and showed despair by exhibiting less mobility time in the FST in the $F_0$ generation. This may be due to the early life exposure to brain insult caused by the prolonged febrile seizure [36]. This finding is in agreement with Crespo et al. [37], who observed that in their model of PFS, rats exhibited depressive-like behavior at adulthood. Furthermore, we also noticed that offspring from both parents with history of PFS consumed less sucrose solution; this may probably be due to genetic transfer and

the level of care received from the dams. This is in agreement with previous study which reported that postpartum depressed mothers had offspring with depressive-like behavior [21]. Interestingly, we observed that there was no change in the offspring immobility time spent within the FST, this may be because loss of hope which is revealed by the FST is one of the late signs of depressive-like behavior [38]. This finding is in accordance with that of Wu et al. [21] who demonstrated that the combined effects of early life maternal separation and maternal stress were only able to increase the immobility time of the offspring in the FST. These depressive-like behaviors has been linked to altered expression of metabotropic receptors (mGluRs) and as such the mGluRs has been suggested as potential targets for drug treatment [39–42]

In this study we observed that mGluR5 mRNA hippocampal expression was down regulated in the PFS group within the $F_0$ generation across all time lines from childhood to adulthood (PND14-60) when compared to the SAL group. This may be due to the active receptor protein bound during excitation subsequently leading to low expression of the unbounded available receptor site in the hippocampus. Our finding is in agreement with the previous report of Iyo et al. [43], who observed low expression of mGluR5 in the hippocampus of their corticosteroid rat model of depression. Among the $F_1$ generation, it was observed in our study that offspring born to both parents with history of PFS and those born of PFSm+SALf parents showed decreased expression of hippocampal mGluR5 mRNA with associated decrease sucrose consumption. This may be due to the gender specific ways by which stress affect gene expression in the hippocampus [44]. Our findings ins in agreement with Wang et al, [40] who in their animal model of depression observed that hippocampal mGluR5 mRNA expression was low in the male offspring when compared to their female counterparts.

Metabotropic glutamate receptor 3(mGluR3) are autoreceptors that decrease excitatory glutamatergic transmission [10]. In this study, we observed that expression of hippocampal mGluR3 was upregulated among the PFS rats when compared to SAL animals at PND14, 37 and 60 in the $F_0$ generation. This is expected since mGluR3 reduces the excitatory effects of glutamate; hence the observed upregulation is due to the active excitatory mGluR5 receptor effect. This finding was corroborated by Stam et al. [45], who reported that in their stressed animals, mGluR3 expression was elevated. Among the $F_1$ generation, it was observed that offspring born of any parent with history of PFS had increased expression of hippocampal mGluR3 mRNA with associated decrease in sucrose consumption when compared to rats born from male and female saline parents. Similar findings was also noticed by Wang et al. [40] who in their prenatal stressed model observed hippocampus of offspring from stressed adult rats and found elevated mGluR3 expression.

Previous studies have demonstrated the involvement of DNA methylation, as a critical epigenetic change in the control of DNA transcription level [46,47]. While there are diverse consequences of DNA methylation on transcription regulation [48, 49], the alteration in DNA methylation level in a distinct region plays a crucial role in the final reaction of DNA methylation consequent to transcription regulation [50]. In this study, we observed the epigenetic effect of prolonged febrile seizure history at different time lines in rats since factors like age, gender, and the environment have been reported to influence DNA methylation [51].

From our methylation study it was observed that methylation of the mGluR1 gene was not of a particular pattern especially at PND14 this was consistent with findings by Bagot et al. [52], where they observed that *Grm1* gene methylation difference which is not specific to a single CpG site. However, it was noticed that PFS resulted more in the hypermethylation of the CpG unit on postnatal day 14, 37 and 60 after the induction of seizure in $F_0$ rats. This pattern was also observed Bagot et al. [52], who observed hypermethylation of *Grm1* gene in offspring of mothers that showed low care (low-LG) when compared to offspring of high-LG mothers. In the $F_1$ generation, animals born to both parents with history of PFS (PFSm+PFSf),showed

increased methylation of mGluR1 when compared to their offspring from both saline parents (SALm+SALf). This may be due to the maternal care and gender specific genetic transfer from parents to offspring. This is in tandem with the findings of Zhang et al. [53], who discovered significant increased methylation level in *GAD1* gene promoter in rat hippocampus in response to maternal care. Furthermore, Wu et al. [21], discovered that environmental factors led to epigenetic changes which bring about susceptibility to disease later on in life. Therefore, we observed that PFS history caused decreased sucrose consumption and immobility in rats in both the sucrose preference test and forced swim test respectively, a concomitant increase in methylation of mGluR1 and mRNA expression of mGluR3 with decreased expression of mGluR5 mRNA in the hippocampus. Our findings were consistent with Wang et al, 2015 [40] who in their chronic stress model observed that offspring of mother exposed to chronic mild stress exhibited depressive behavior, decreased mGluR5 hippocampal expression and increased mGluR2/3 expression. Our present observation was also in agreement with Bagot et al, 2012 [14] who showed that poor maternal care which could be as a result of early life time event in rats can lead to behavioral changes and hypermethylation of mGluR1 hippocampal expression in their offspring.

We concluded that PFS may profoundly affect mGluR1 methylation profile, mRNA expression of both mGluR5 and mGluR3 genes which may have led to depressive-like behavior in rats in both the $F_0$ and $F_1$ generations. Thus, the metabotropic receptor proteins could be a novel target in drug development towards the treatment of major depression.

## Supporting information

**S1 Table. Showing the synthesis reports of mGluR5 (Grm5) and mGluR3 (Grm3) primers used to assay genes in the hippocampal tissue in this stud.**
(PDF)

**S2 Table. Showing the raw data generated from the Mass ARRAY analysis of mGluR1 gene methylation status used in this study.**
(CSV)

## Acknowledgments

The authors would like to appreciate the staff of the Biomedical Resource Centre of the University of KwaZulu-Natal for the technical assistance provided.

## Author Contributions

**Conceptualization:** Oluwole Ojo Alese, Musa V. Mabandla.

**Data curation:** Oluwole Ojo Alese.

**Formal analysis:** Oluwole Ojo Alese.

**Funding acquisition:** Musa V. Mabandla.

**Investigation:** Oluwole Ojo Alese.

**Methodology:** Oluwole Ojo Alese.

**Resources:** Musa V. Mabandla.

**Supervision:** Musa V. Mabandla.

**Validation:** Musa V. Mabandla.

**Writing – original draft:** Oluwole Ojo Alese.

**Writing – review & editing:** Oluwole Ojo Alese, Musa V. Mabandla.

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
