## [Decision Letter · Decision Letter 0]

11 Oct 2019

PONE-D-19-24426

Transgenerational deep sequencing revealed hypermethylation of hippocampal mGluR1 gene with altered mRNA expression of mGluR5 and mGluR3 associated with behavioral changes in Sprague Dawley rats with history of prolonged febrile seizure

PLOS ONE

Dear Dr Alese,

Thank you for submitting your manuscript to PLOS ONE. After careful consideration, we feel that it has merit but does not fully meet PLOS ONE’s publication criteria as it currently stands. Therefore, we invite you to submit a revised version of the manuscript that addresses the points raised during the review process. The reviewer comments can be found below.

We would appreciate receiving your revised manuscript by Nov 25 2019 11:59PM. To enhance the reproducibility of your results, we recommend that if applicable you deposit your laboratory protocols in protocols.io, where a protocol can be assigned its own identifier (DOI) such that it can be cited independently in the future. For instructions see: http://journals.plos.org/plosone/s/submission-guidelines#loc-laboratory-protocols

We look forward to receiving your revised manuscript.

Kind regards,

Judith Homberg

Academic Editor

PLOS ONE

**Journal Requirements:**

**Comments to the Author**

1. Is the manuscript technically sound, and do the data support the conclusions?

Reviewer #1: No

2. Has the statistical analysis been performed appropriately and rigorously? 

Reviewer #1: Yes

3. Have the authors made all data underlying the findings in their manuscript fully available?

Reviewer #1: No

4. Is the manuscript presented in an intelligible fashion and written in standard English?

Reviewer #1: Yes

5. Review Comments to the Author

Reviewer #1: Dr Oluwole Alese and collaborators present their work on inheritable behavioral changes after experimental febrile seizures that are possibly involved in hypermethylation of hippocampal mGluR1 gene and altered mRNA expression of mGluR5 and mGluR3. The topic is interesting. However, there are a number of points that deserve the authors' attention:

1. The model of PFS should be verified by EEG recording of hippocampus throughout

the experiment.

2. The correlation between behavioral changes and hypermethylation of hippocampal

mGluR1 gene with altered mRNA expression of mGluR5 and mGluR3 is lack of evidence，some interventions, like inhibitors or agonists were needed.

3. Why the behavioral changes in F1 generation as shown in FIG.2C were not

matched with the mRNA expression of mGluR5 and mGluR3 as shown in FIG.4D and 5D，especially the offspring of PFS-M and SAL-F？

4. As mGluR1 was hypermethylated after PFS in F0 and F1 generation, what about

its mRNA expression?

6. PLOS authors have the option to publish the peer review history of their article (what does this mean?). If published, this will include your full peer review and any attached files.

Reviewer #1: No

---

## [Author Response · Author response to Decision Letter 0]

16 Oct 2019

1. Is the manuscript technically sound, and do the data support the conclusions?

Reviewer #1: No 

We appreciate your comments the technical concerns has been improved upon and the manuscript revised accordingly.

2. Has the statistical analysis been performed appropriately and rigorously? 

Reviewer #1: Yes 

Thank you for your commendation. 

3. Have the authors made all data underlying the findings in their manuscript fully available?

Reviewer #1: No 

As per the valuable reviewer we have included more of the raw data and data sheets as part of the supporting information.

4. Is the manuscript presented in an intelligible fashion and written in standard English?

Reviewer #1: Yes 

Thank you for your commendation. 

5. The model of PFS should be verified by EEG recording of hippocampus throughout the experiment.

As per the valuable reviewer comment, the model was first used and validated by Heida et al in 2005 which had been reproduced in our laboratory (Cassim et al., 2015; Qulu et al., 2016; Mkhize et al., 2017; Rakgantsho and Mabandla, 2019; Alese and mabandla, 2019). LPS is a component of gram negative bacteria cell wall which is known to reliably induce fever in rodent (Ostberg et al., 2000: Cooper et al., 1964). LPS at a dose of 200µg/kg or more is known to cause fever in rats and also addition of kianic acid (KA) will further increase the body temperature and consequently leads to seizure in the pups (Heida et al., 2005). The EEG recording may not provide an additional information since this is an already established model by Heida et al., 2005 and EEG has been previously used to confirm LPS/KA induced seizure. As indicated by our finding that “All rats induced had at least stage 4 convulsions for a minimum of 30min duration” indicated that the convulsion were visibly obvious to the observer as rearing with bilateral forelimb clonus which is consistent with earlier observation (Racine, R.J., 1972. Modification of seizure activity by electrical stimulation: II. Motor seizure)(page 5, paragraph 2, lines 11-14). 

6. The correlation between behavioral changes and hypermethylation of hippocampal mGluR1 gene with altered mRNA expression of mGluR5 and mGluR3 is lack of evidence，some interventions, like inhibitors or agonists were needed. 

Regarding the valuable reviewer recommendation the following statement has been included to correlate our findings in this study. “Therefore, we observed that PFS history caused decreased sucrose consumption and immobility in rats in both the sucrose preference test and forced swim test respectively, a concomitant increase in methylation of mGluR1 and mRNA expression of mGluR3 with decreased expression of mGluR5 mRNA in the hippocampus. Our findings were consistent with Wang et al, 2015 who in their chronic stress model observed that offspring of mother exposed to chronic mild stress exhibited depressive behavior, decreased mGluR5 hippocampal expression and increased mGluR2/3 expression. Our present observation was also in agreement with Bagot et al, 2012 who showed that poor maternal care which could be as a result of early life time event in rats can lead to behavioral changes and hypermethylation of mGluR1 hippocampal expression in their offspring”. (page 18 line 1-10)

7. Why the behavioral changes in F1 generation as shown in FIG.2C were not matched with the mRNA expression of mGluR5 and mGluR3 as shown in FIG.4D and 5D，especially the offspring of PFS-M and SAL-F？

Thank you for your valuable comments, the statement “Among the F1 generation, it was observed in our study that offspring born to both parents with history of PFS and those born of PFSm+SALf parents showed decreased expression of hippocampal mGluR5 mRNA” has been changed to “Among the F1 generation, it was observed in our study that offspring born to both parents with history of PFS and those born of PFSm+SALf parents showed decreased expression of hippocampal mGluR5 mRNA with associated decrease sucrose consumption”. This is to bring more clarity to the discussion (page 16 lines 4-7). Also the following statement “Among the F1 generation, it was observed that offspring born of any parent with history of PFS had increased expression of hippocampal mGluR3 mRNA when compared to rats born from male and female saline parents” has been changed to “Among the F1 generation, it was observed that offspring born of any parent with history of PFS had increased expression of hippocampal mGluR3 mRNA with associated decrease in sucrose consumption when compared to rats born from male and female saline parents” this is to bring more clarity to the discussion (page 16, paragraph 2, lines 7-10).

8. As mGluR1 was hypermethylated after PFS in F0 and F1 generation, what about its mRNA expression? 

Our focus in this study was on the epigenetic variation of mGluR1 and mRNA expression of mGluR3 and mGluR5. Since mGluR1 and mGluR5 are subtypes of group 1 metabotropic glutamate receptor, they may therefore have a modulatory/ synergistic effect on each other and thus may show the same pattern of expression (Bonsi et al., 2005; Rae and Irving 2004). In the light of this we decided to carry out different assays on the 2 receptors of same group in our study to avoid duplication of techniques.

---

## [Decision Letter · Decision Letter 1]

29 Oct 2019

Transgenerational deep sequencing revealed hypermethylation of hippocampal mGluR1 gene with altered mRNA expression of mGluR5 and mGluR3 associated with behavioral changes in Sprague Dawley rats with history of prolonged febrile seizure

PONE-D-19-24426R1

Dear Dr. Alese,

We are pleased to inform you that your manuscript has been judged scientifically suitable for publication and will be formally accepted for publication once it complies with all outstanding technical requirements.

With kind regards,

Judith Homberg

Academic Editor

PLOS ONE

Additional Editor Comments (optional):

Reviewers' comments:

Reviewer's Responses to Questions

**Comments to the Author**

1. If the authors have adequately addressed your comments raised in a previous round of review and you feel that this manuscript is now acceptable for publication, you may indicate that here to bypass the “Comments to the Author” section, enter your conflict of interest statement in the “Confidential to Editor” section, and submit your "Accept" recommendation.

Reviewer #1: All comments have been addressed

2. Is the manuscript technically sound, and do the data support the conclusions?

Reviewer #1: Yes

3. Has the statistical analysis been performed appropriately and rigorously? 

Reviewer #1: Yes

4. Have the authors made all data underlying the findings in their manuscript fully available?

Reviewer #1: (No Response)

5. Is the manuscript presented in an intelligible fashion and written in standard English?

Reviewer #1: Yes

6. Review Comments to the Author

Reviewer #1: (No Response)

7. PLOS authors have the option to publish the peer review history of their article (what does this mean?). If published, this will include your full peer review and any attached files.

Reviewer #1: No

---

## [Editor Report · Acceptance letter]

31 Oct 2019

PONE-D-19-24426R1 

Transgenerational deep sequencing revealed hypermethylation of hippocampal mGluR1 gene with altered mRNA expression of mGluR5 and mGluR3 associated with behavioral changes in Sprague Dawley rats with history of prolonged febrile seizure 

Dear Dr. Alese:

I am pleased to inform you that your manuscript has been deemed suitable for publication in PLOS ONE. Congratulations! Your manuscript is now with our production department. 

With kind regards,

on behalf of

Dr. Judith Homberg 

Academic Editor

PLOS ONE